# Evaluation of a Novel Technique for Closure of Small Palatal Fistula

**DOI:** 10.3390/jpm13010065

**Published:** 2022-12-28

**Authors:** Edoardo Brauner, Luca Piccoli, Karim Sallemi, Umberto Romeo, Federico Laudoni, Marco Cantore, Gianluca Tenore, Nicola Pranno, Francesca De Angelis, Michele Di Cosola, Valentino Valentini, Stefano Di Carlo

**Affiliations:** 1Department of Oral and Maxillofacial Sciences, Sapienza University of Rome, Via Caserta 6, 00161 Rome, Italy; 2Implanto-Prosthetic, Policlinico Umberto I, Viale Regina Elena 287b, 00161 Rome, Italy; 3Department of Clinical and Experimental Medicine, University of Foggia, 71122 Foggia, Italy; 4Oncological and Reconstructive Maxillo-Facial Surgery Unit, Policlinico Umberto I, Viale del Policlinico 155, 00167 Rome, Italy

**Keywords:** oral surgery, palatal fistula, minimally invasive

## Abstract

A palatal fistula is a pathological condition that connects the nasal cavities with the oral cavity. An oral–nasal fistula is reported as a possible post-surgical complication after the removal of oral carcinomas. The presence of a palatal fistula affects the patient’s quality of life, making it necessary to apply a prosthetic device, such as a palatal plate, to keep the nasal cavities separated from the oral one. There are several surgical techniques to close a palatal fistula, but it is not possible to define the optimal technique as the approach is extremely dependent on the characteristics of the fistula. The aim of this article is to propose a minimally invasive technique to reduce the size of palatal fistulae and to reduce the surgical difficulty (NSPF). A total of 20 patients fulfilled the inclusion criteria and were checked every two weeks. The fistula was injured with a needle every 2 weeks. Fifteen patients who healed with complete closure of the fistula reported no need for a palatal protection plate to eat, drink and speak normally. It is possible to conclude that the NSPF protocol is a valid approach for the non-surgical reduction of palatal fistulae, and it is possible, when the appropriate conditions are present, to achieve complete closure.

## 1. Introduction

Palatine fistula is an anatomical condition that connects the oral cavity with the nasal cavities. The size and depth of extension depends on several factors. This can be a complication of various underlying pathologies, such as oral carcinoma, or it can be caused by intra-operative accidents during oral surgery, such as extractions. Palatal fistulae are often characterized by a duct covered by a self-secreting epithelium, whose spontaneous closure is extremely difficult and is complicated by local factors in the oral cavity, such as the presence of bacterial contamination and continuous trauma due to chewing and breathing. A patient with a palatal fistula often has difficulty in swallowing, speaking and chewing. Furthermore, communication with the nasal cavities can lead to bacterial superinfections, with the consequent development of sinusitis, affecting the paranasal sinuses. Clinical evaluation of the fistula is mainly based on the location of the fistula based on the Pittsburg classification (Figure 1).

Many patients who require resective surgery for maxillo-facial tumors often present functional problems, such as a limited ability to swallow, altered phonetics and difficulty in chewing. Among the various possible therapies, patients who receive surgery often require prosthetic rehabilitations. One of the post-surgical complications is represented by the palatal fistula, an anatomical condition that requires the use of prosthetic devices such as palatal protection plates.

Protection plates are a viable solution in improving patient functionality, but they remain a removable solution, not improving the patient’s quality of life [2,3,4,5].

However, Kornblith shows, in his cross-sectional study, how the size of the palatal resection and the relative size of the palatal protection plate can strongly influence the psychological, sexual and family wellbeing of the patient, making surgery a better therapeutic solution than removable prostheses [4].

There are several surgical techniques to close palatal fistulae, but none is superior as it depends on the patient’s subjective characteristics (age and surgical history), the characteristics of the fistula (size and location) and the surgeon’s experience [3].

The clinical cases that we discuss in this article involve localized fistulae in the hard palate, the hard/soft palate junction and the soft palate according to the Pittsburg classification [6] (Figure 1).

Considering the discomfort that a protection plate can cause to the patient, and the different prognoses of surgery for the closure of palatal fistulae, we consider it necessary to devise a minimally invasive and predictable approach to improve and, when possible, completely solve the problem [6,7,8,9,10]. We therefore introduce a minimally invasive non-surgical approach with the aim to identify which factors are fundamental to increase the likelihood of reducing or completely closing the palatal fistula (NSPF).

The difficulty of the spontaneous closure of a palatal fistula is intrinsic in its pathogenesis; the fistula is generally described as a pathological epithelial duct that connects anatomical areas that normally do not communicate. It is therefore possible to define a palatal fistula as a duct that forms at least 48 h after the formation of communication between the nose and the oral cavity, as it is possible to observe the epithelialization of the limits of the wound.

In palatal fistulae, communication occurs between the nasal and paranasal structures and the oral cavity. In the literature, it is mainly described as a complication of therapies performed on the upper jaw. Extractions of dental elements with apices that present contiguity with the maxillary sinus, tuber fractures and implant therapies associated with sinus lifts can result in oro-antral communication, which, if not healed in 48 h, leads to epithelialization.

Palatine fistulae can also occur following firearm accidents; however, they are poorly described in the literature, probably due to the high mortality rate [5,6,9,11,12].

Palatal fistulae can be a secondary complication to bacterial infection not necessarily linked to the oral cavity. Güzel MZ et al. describe a palatal fistula as a secondary complication to tuberculous dacryoadenitis infection, a very rare but possible situation [5,6,11,12,13].

Palatine fistula therapies can be divided into two large groups: surgical therapy and non-surgical therapy.

The main approach to palatal fistulae is surgical therapy. The choice of the surgical technique is mainly made based on the location and size [14,15,16]. Hu et al. describe a very detailed decision tree (Figure 2) based on the Pittsburg classification (Figure 1) [17]. The timing and the choice of the surgical technique allow an increase in the success rate in closing the palatal fistulae [18].

The only non-surgical approach that can be found in the literature has been described by Berkman et al. In 1978, Berkman performed palatal protection on 11 patients approximately 10 days after the detection of the fistula [10]. All patients had a palatal fistula located in the hard palate, the hard/soft palate junction or the soft palate after a palato-plasty and/or a pharyngeal flap.

According to the authors, the selection of surgical therapy is crucial, as it allows for a minimally invasive approach [15]. However, it can lead to the onset of post-operative complications, such as, in addition to fistulae, OSA. However, there are no articles on non-surgical therapies in the literature. The authors observed that untreated fistulae tended to contract spontaneously before stabilizing. Therefore, we asked whether it was possible, applying palatal protection and keeping the fistula bleeding, to stop the contraction process.

The aim of this article was therefore to propose a non-surgical approach to close palatal fistulae.

## 2. Materials and Methods

A total of 47 patients were rehabilitated in our head and neck integrated care department, in the complex operating unit of implantology and dental prosthetics, at the Department of Oral Sciences and Maxillofacial Surgery, Policlinico Umberto I, Sapienza, University of Rome (Rome, Italy).

Patients with primary oral cancers were included. In accordance with the Pittsburg Classification, patients with a fistula at the level of the hard palate, soft palate, soft/hard palate junction and labio-alveolar junction or the uvula were included. Fifteen patients who could not be monitored every 2 weeks were excluded. Five patients were found to be substance abusers (cocaine) and were therefore excluded. Seven patients with osteonecrosis of the maxilla were excluded.

Therefore, only 20 patients were considered in our evaluation (Table 1). All 20 patients considered for the study wished to find an alternative to surgery.

All the patients were initially rehabilitated with a palatal protection plate composed of the same material (resin) and produced by the same dental laboratory. In some patients, who were totally or partially dentulous, it was necessary to perform palatal protection that included an orthopedic prosthesis. Depending on the aesthetic or functional needs of the patient, the color of the material could vary but the chemical composition was the same.

All the patients underwent surgery that resulted in a fistula between the oral cavity and the nasal cavity.

Size was clinically measured by the same operator with the PCP15 (Iso) and photographs taken at each follow-up appointment. The PCP15 probe is a millimeter-scale periodontal probe with sections spaced at every mm. The size of the fistula was measured in the anteroposterior direction for the length, and the width was measured in the disto-mesial and mesio-distal directions with respect to the median line. During each appointment, a small wound was created around the fistula edges to induce bleeding. In the same session, the palatal plate was reduced by approximately 0.5 mm in the laboratory. Patients were treated and monitored every 2 weeks, according to the principles of the healing processes of the gum. At each appointment, a photograph was taken before treatment.

In order to critically analyze the results, the patients were divided into two groups:

Group 1 patients who had at least one dental element;

Group 2: patients with total edentulism.

The two groups were further divided into two subgroups:

Subgroup A: patients with fistulae involving the maxillary bone;

Subgroup B: patients with fistulae without maxillary bone involvement (Table 2).

The patients were also subdivided according to the Pittsburgh classification (Table 3).

All patients were treated with NSPF.

## 3. Results

Twenty patients out of 47 treated at the department met the inclusion criteria. These 20 patients (12 females and 8 males) were checked every two weeks. The patient’s average age was 70, with a minimum age of 19 and a maximum age of 90.

The healing process generally took place in two phases: an initial one characterized by a greater tissue response, with the more rapid closure of the palatal fistula, and a subsequent slower and more gradual one. On average, the greater and faster healing phase was observed in the first 2 weeks of treatment, with 50–60% fistula obliteration. Of the 20 patients involved, healing with restitutio ad integrum was achieved in 15 cases, while an average reduction on average of 5 mm was obtained in the remaining patients. Eleven patients belonged to group 1 and four patients belonged to group 2 (Table 4).

We also observed that 11 of the patients with 100% fistula healing belonged to sub-group A. Below, we present some of the clinical cases treated.

### 3.1. Case Reports

#### 3.1.1. Case 1: A.R.

A 90-year-old female patient was referred from the maxillofacial surgery unit. The patient underwent a right hemi-maxillectomy for maxillary carcinoma. The patient’s history was negative for any type of systemic or local pathology. The patient had a type III palatal fistula based on the Pittsburg classification (Figure 3).

When the protective plate was applied, three types of mucosae were observed around the fistula: alveolar mucosa, soft palate posteriorly, and keratinized gingiva on the hard palate.

The protective plate was composed of resin and consisted of a dental and mucous support (Figure 4 and Figure 5).

Every 2 weeks, the edges of the fistula were gently injured with the point of a needle and the palatal protection plate was then repositioned. No drug therapy was administered, and the patient continued with their normal oral hygiene habits.

After one month, we clinically observed a reduction in the diameter of the fistula and the patient reported an improvement in chewing and swallowing (Figure 6 and Figure 7).

#### 3.1.2. Case 2: P.S.

A 19-year-old female patient with a previous palatal carcinoma. The patient was being treated with Metformin for type 2 diabetes. She reported only an allergy to penicillin. She was referred to our department after the removal of a type II palatal fistula carcinoma according to the Pittsburg classification (Figure 8). The patient had all the dental elements of the upper arch, and the basal maxillary bone was not involved, so she belonged to group 1, subgroup B, according to the classification in Table 2.

The patient was then subjected to the NSPF protocol. (Figure 9, Figure 10 and Figure 11) It was possible to observe a progressive reduction in size until the complete closure of the fistula was achieved. (Figure 12)

#### 3.1.3. Case 3: R.M.

A 58-year-old patient with a type VI maxillary fistula without involvement of the maxillary bone. Therefore, the patient was classified as group 1, subgroup B. Negative history of any pathology or drug therapy. The patient had a fistula in the right upper vestibular fornix, with disto-mesial extension of approximately 3 cm (Figure 13). The patient presented with a previous upper partial denture. A new bilateral resin partial prosthesis (Figure 14) with an extension inside the fistula was planned together with the NSPF protocol. After two months of therapy, a significant reduction in the size of the fistula is observed. (Figure 15).

After two months, the complete closure of the fistula was observed (Figure 14).

#### 3.1.4. Case 4: P.M.A.

A 70-year-old patient presented after surgical resection of a palatal carcinoma caused by lichen planus. The patient was taking beta-blocking medication. The patient was edentulous in the upper and lower jaw, requiring total prosthetic rehabilitation with the addition of resin to fill the fistula. (Figure 16) The patient’s fistula could be classified as type III according to the Pittsburg classification and was placed in group 2, subgroup B, as the basal bone of the maxilla was involved.

The patient was treated with NSPF every two weeks, with a progressive reduction in the portion of the prosthesis that was introduced into the fistula. The fistula was initially surrounded by hypertrophic fibro mucosae. After one month, there was a marked improvement in the quality of the tissue around the fistula, but the size was not reduced as expected. After two months, there was no change in size, with the exception of a reduction in the reported symptoms and an improvement in the mucosa around the fistula. (Figure 17).

## 4. Discussion

The main objective of the study was to propose a minimally invasive technique to reduce the size of palatal fistulae and to reduce the surgical difficulty.

The significant reduction in size allowed us to simplify the surgical procedure to close the fistula. In some cases, the NSPF protocol led to restitutio ad integrum (15 patients), allowing the complete recovery of function.

It is possible that the bleeding induced cell proliferation and therefore a constant reduction in the distance between the walls of the fistula with healing modalities by secondary intention.

This possibly occurred because the patients were younger, meaning that cell replication took place more rapidly and efficiently.

The presence of at least one dental element ensured the greater stability of the prosthesis, with the advantage of less trauma in the fistula region during stomatognathic functions.

Similarly, the presence of maxillary bone was also a favorable element for tissue healing.

On the other hand, the involvement of the maxillary bone by the primary tumor and total edentulism were found to be unfavorable factors. In such cases, in fact, an increase in the mobility of the prosthesis was recorded during periodic checks, causing constant trauma to the fistula and a reduced tissue response.

In addition, the presence of keratinized tissue in the lesion, by virtue of its increased consistency, ensured, in the cases concerned [10], better resistance to trauma and cell proliferation, after the bleeding and the obliteration of the fistula with the prosthesis.

In fact, another favorable factor for healing is the reduction of bacterial colonization. In this sense, not only was the prosthesis–mucosa seal decisive, being periodically re-evaluated at each check-up, but also the patient’s compliance was ensured in terms of the correct at-home hygiene practices. Although we observed a high rate of complete closure of the treated fistulae (75%), it must be considered that the majority of the treated patients had type III and IV fistulae (Table 3). Type III and IV fistulae, according to the Pittsburg classification, do not have muscle-type mobility. Moreover, the stabilization of the palatal protection is easier to obtain.

## 5. Conclusions

In conclusion, we cannot prove the particular effectiveness of the NSPF protocol, as a more evidence-based research methodology and more control studies are required. We can hypothesize the significant importance of non-surgical techniques. The conditions that increase the likelihood of closing the fistulae, according to what has been clinically observed, are as follows.

Age: Regenerative capacity at a young age is likely to be greater.The presence of dental elements: A greater reduction in size and a higher rate of closure was observed in partially edentulous patients than in totally edentulous patients. Dental stabilization of the palatal plate probably reduces any micro-movements that prevent the apposition of tissue.Presence of basal maxillary bone: The presence of basal bone seems to improve the quantity and quality of the maturation tissue, probably acting as a support for the newly formed tissue.The presence of keratinized mucosa: The resistance to trauma of the keratinized epithelium of the mucosa improves the stability over time of the result obtained from the reduction in the diameter of the fistula. The palatal fibromucosa is extremely resistant to trauma compared to the alveolar mucosa [19].

A multicenter study would allow us to achieve a statistically valid sample number and greater diversification regarding the typology of fistulae according to the Pittsburg classification. Further evaluation will be needed to understand the true potential of this technique. It appears to be valid for the purpose of reducing large fistulae and reducing surgical invasiveness. It will certainly be necessary to observe patients to understand the stability of the result obtained over time, so as to define guidelines and protocols with predictable prognoses.

## Figures and Tables

**Figure 1 jpm-13-00065-f001:**
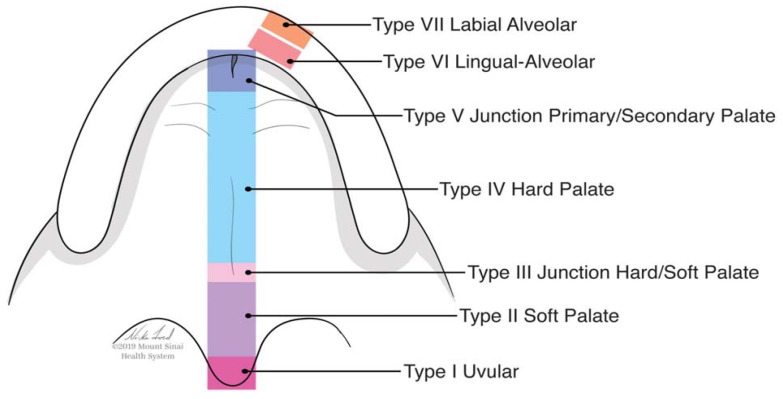
The Pittsburgh Fistula Classification System for palatal fistulae (Adapted with permission from [1]. Copyright year 2007, Darren M Smith).

**Figure 2 jpm-13-00065-f002:**
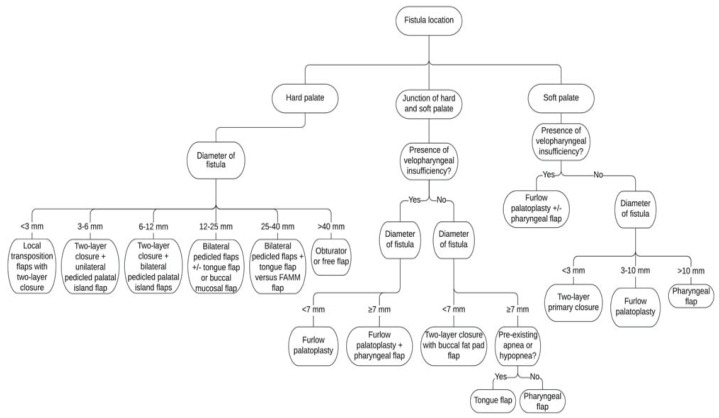
Decision tree on surgical techniques for palatal fistulae (Reprinted with permission from [17]. Copyright 2020, Shirley Hu.)

**Figure 3 jpm-13-00065-f003:**
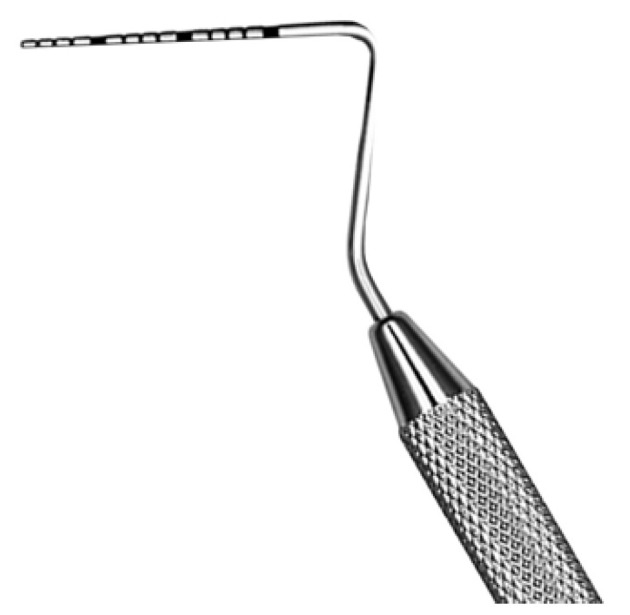
PCP15 periodontal probe.

**Figure 4 jpm-13-00065-f004:**
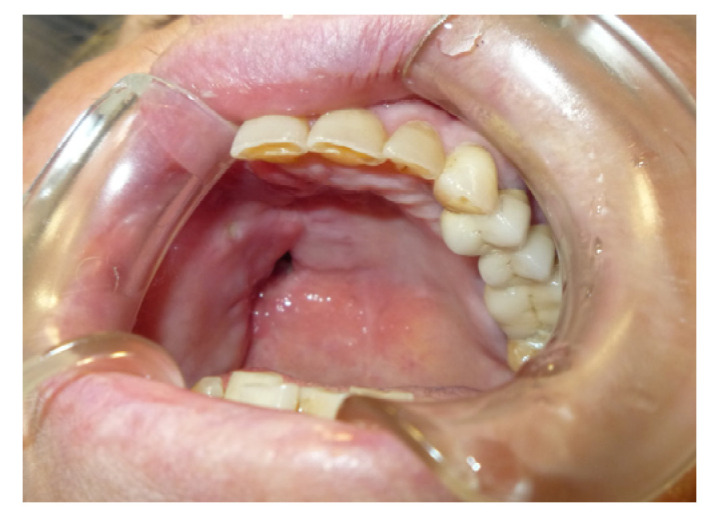
Initial application of the protection plate.

**Figure 5 jpm-13-00065-f005:**
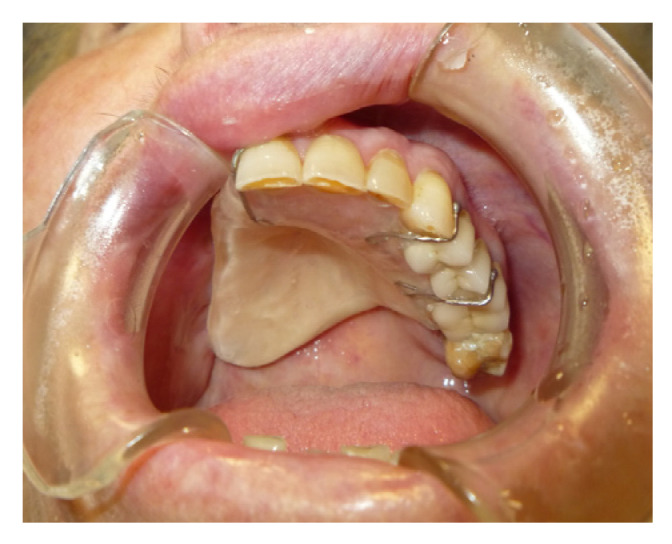
Palatal plate.

**Figure 6 jpm-13-00065-f006:**
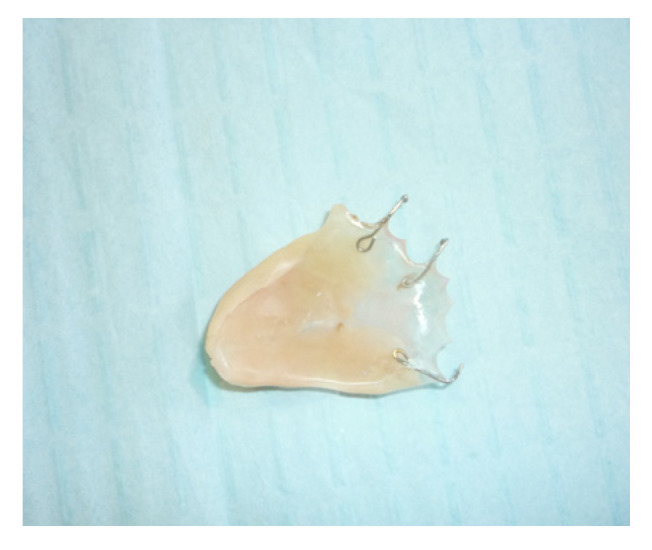
Palatal plate detail.

**Figure 7 jpm-13-00065-f007:**
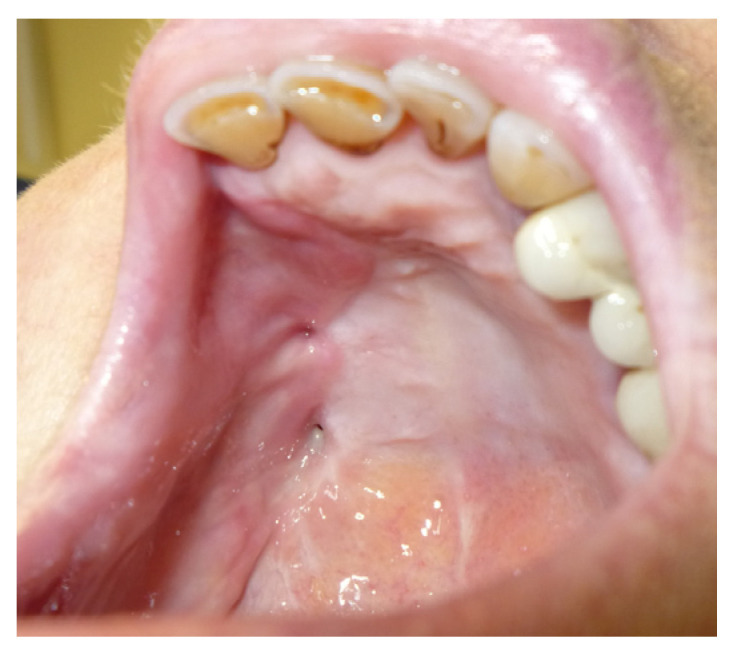
After one month.

**Figure 8 jpm-13-00065-f008:**
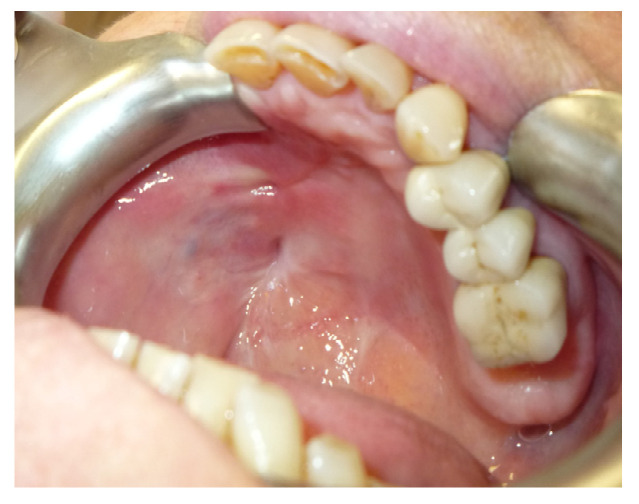
After 2 months.

**Figure 9 jpm-13-00065-f009:**
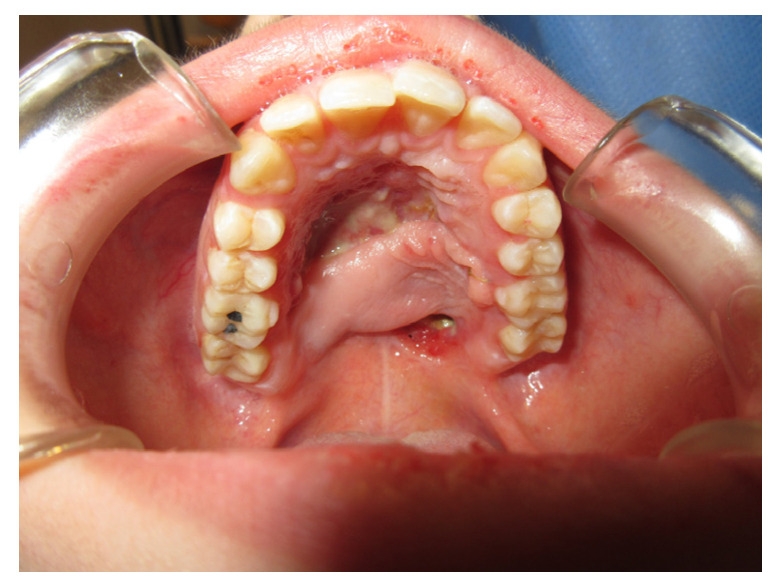
Day one before the application of the palatal plate.

**Figure 10 jpm-13-00065-f010:**
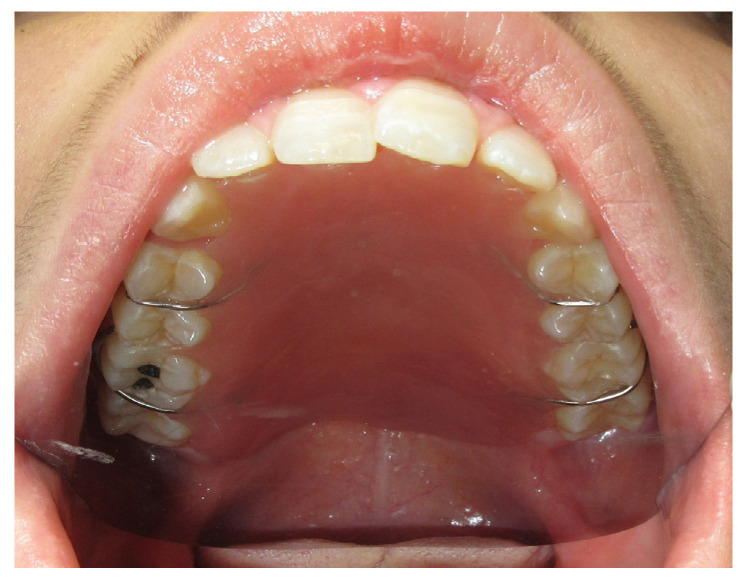
Palatal plate in position.

**Figure 11 jpm-13-00065-f011:**
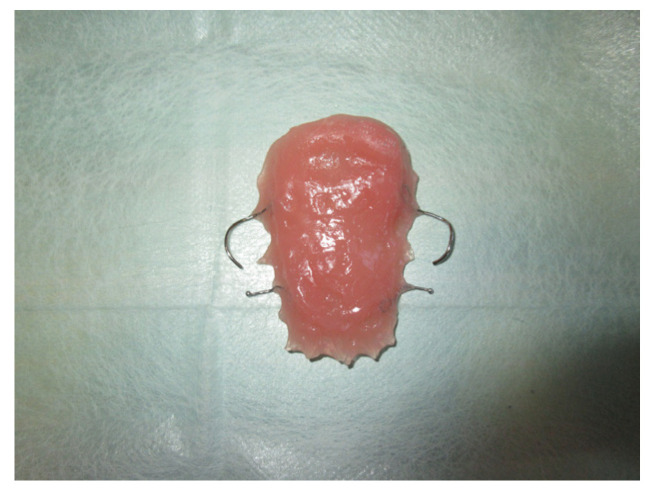
Extraoral vision of the palatal plate.

**Figure 12 jpm-13-00065-f012:**
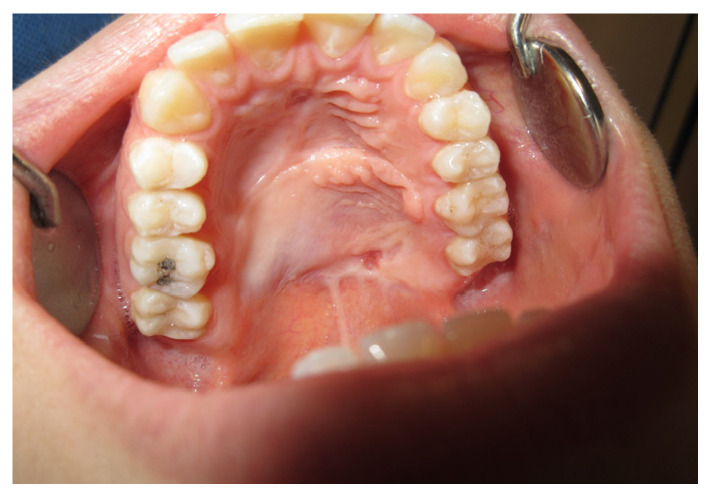
Intra-oral vision after 2 months.

**Figure 13 jpm-13-00065-f013:**
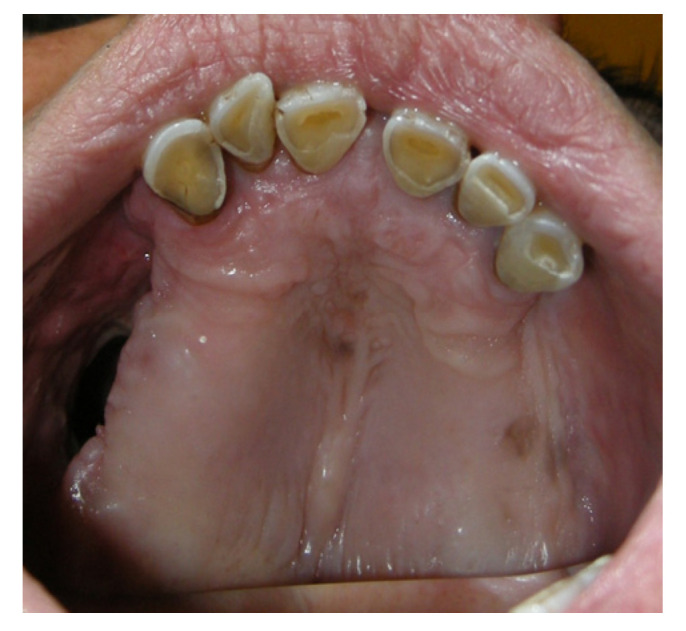
Patient condition at the first visit.

**Figure 14 jpm-13-00065-f014:**
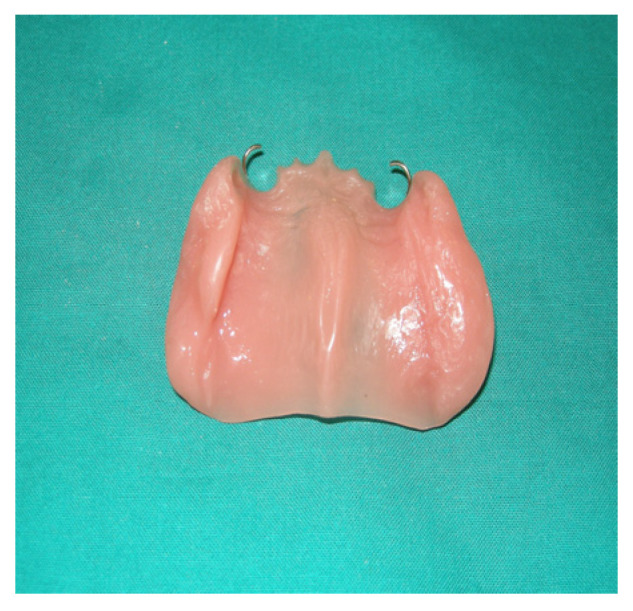
Partial prosthesis in resin with function of palatal plate.

**Figure 15 jpm-13-00065-f015:**
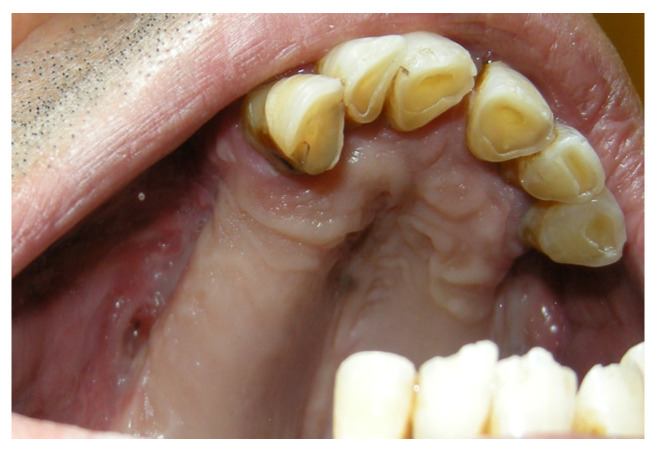
After two months.

**Figure 16 jpm-13-00065-f016:**
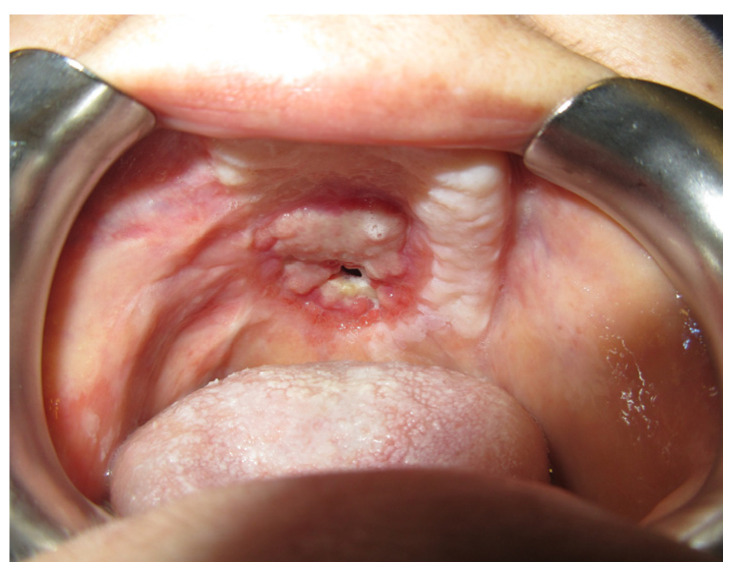
Before NSPF.

**Figure 17 jpm-13-00065-f017:**
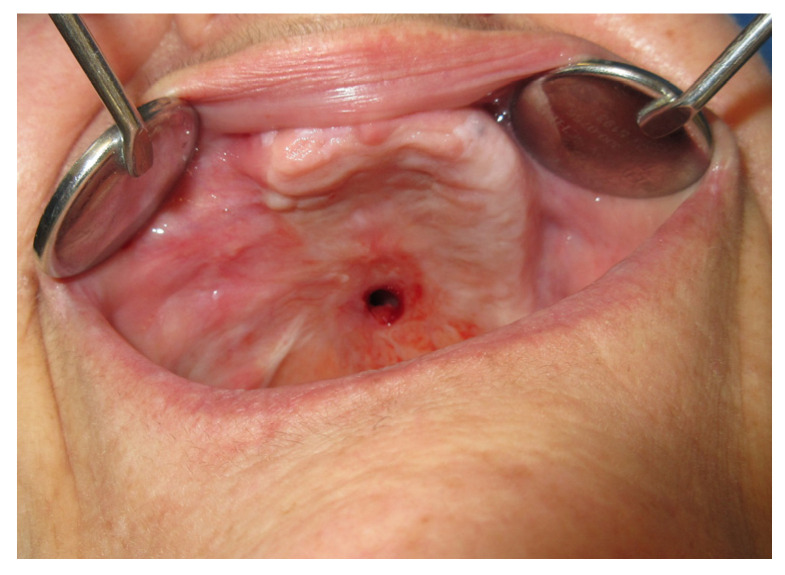
After 2 months.

**Table 1 jpm-13-00065-t001:** Evaluation of 20 patients with primary oral cancers.

Name	Gender	Diagnosis	Therapy	Prosthetic Rehabilitation
A.R	F	K superior jaw	Surgery	Superior
C.G.	F	Adenocarcinoma	Surgery	Superior
C.A.	F	Pleomorphic adenoma left hemimaxilla	Surgery	Superior + inferior
D.M.	M	K Adenocystic salivary glands of the palate	Surgery + Rt	Superior
D.A.	F	Chondrosarcoma nasal bones	Surgery	Superior
D.V.	M	K Squamous cell soft palate	Surgery	Superior
G.P.	F	Palate adenocarcinoma	Surgery	Superior
I.C.	F	K of the palate	Surgery	Superior
L.G.	F	K Squamous cell maxilla	Surgery	Superior + inferior
M.R.	M	K Mucoepidermoid maxilla	Surgery	Superior
M.R.2	F	K of the maxilla	Surgery	Superior
P.S.	F	K of the palate	Surgery	Superior
S.M.	M	K Right cheek and right retromolar spacer	Surgery + Rt adjuvant	Superior + inferior
S.L.	F	Chondroblastic osteosarcoma maxilla	Surgery	Superior
T.E.	M	Osteosarcoma maxilla	CHT neoadjuvant + surgery + CHT adjuvant	Superior
T.G.	F	K Adenocystic maxilla	Surgery	Superior
V.F.	M	K Right retromolar space	Surgery	Superior
R.D.	M	K Jaw	Surgery	Superior
R.M.	M	K Squamous cell maxilla	Surgery	Superior
L.M	F	K Squamous cell soft palate	Surgery	Superior

K—Carcinoma; Rt—Radiotherapy; CHT—Chemotherapy.

**Table 2 jpm-13-00065-t002:** Patients with fistulae without maxillary bone involvement.

	Subgroup A	Subgroup B	Total
Group 1	7	6	13
Group 2	4	3	7

**Table 3 jpm-13-00065-t003:** Pittsburgh classification.

Pittsburg Classification	N° Patients
Type I	0
Type II	3
Type III	6
Type IV	10
Type V	0
Type VI	1
Type VII	0

**Table 4 jpm-13-00065-t004:** Sizes of the fistulae.

N°	Group 1	Group 2	Subgroup A	Subgroup B	Initial Size (mm)	After NPSF (mm)
A.R	X		*		5	1
C.G.		X	*		6	0
C.A.	X			*	4	0
D.M.	X			*	5	0
D.A.	X		*		7	0
D.V.		X		*	7	2
G.P.	X		*		8	0
I.C.	X		*		7	0
L.G.		X	*		8	0
M.R.	X			*	3	0
M.R.2		X		*	4	3
P.S.	X		*		7	0
S.M.	X			*	4	0
S.L.	X		*		6	2
T.E.		X	*		2	0
T.G.	X		*		5	1
V.F.		X	*		3	0
R.D.	X			*	6	0
R.M.	X			*	9	0
P.M.A		X		*	5	5

X—indication of the group; *—indication of the subgroup.

## Data Availability

Not applicable.

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
