# Peer review of "Evaluation of a Novel Technique for Closure of Small Palatal Fistula"

_jpm, 2022, doi:10.3390/jpm13010065_

Round 1
Reviewer 1 Report (New Reviewer)
After having read your paper I think the title should be changed by adding "... palatal fistula closure in non cleft patients" since most readers would expect a study on the closure of cleft fistulae.
This is the description of a treatment attempt for a very limited number of patients who may have contra-indications to surgery, do not want to be operated or where several attempts to fistula closure have failed.
As you correctly state in your conclusion your paper may serve as a base for a study with a much larger patient number. I would suggest to start a multicenter study on this behalf and publish these results in the end.
You mentioned the measurement of the fistulae by with PCP15(iso). Please explain this in detail.
There are several spelling or grammar errors (line 91, 97, 110,144 ,240,253-254.
Finally fig. 9 partially overlaps the text, and fig.10 is not marked .
Author Response
Point 1: After having read your paper I think the title should be changed by adding "... palatal fistula closure in non cleft patients" since most readers would expect a study on the closure of cleft fistulae.
Response 1: we decided to change the title as you recommended. Thanks for your kind reply
Point 2: This is the description of a treatment attempt for a very limited number of patients who may have contra-indications to surgery, do not want to be operated or where several attempts to fistula closure have failed.
Response 2: the patients required an alternative to surgery, none of the patients had cons-indications.
Point 3:As you correctly state in your conclusion your paper may serve as a base for a study with a much larger patient number. I would suggest to start a multicenter study on this behalf and publish these results in the end.
Response 3:
we agree on the fact that a multicenter study can give a greater sample size and variability in the localization of the fistulas
Point 4: You mentioned the measurement of the fistulae by with PCP15(iso). Please explain this in detail.
Response 4:
The pcp 15 is a dental instrument typically used in periodontology. we have added a small description to help you understand.
Point 5:There are several spelling or grammar errors (line 91, 97, 110,144 ,240,253-254.
Response 5:
thank you very much, we corrected them.
Point 6: Finally fig. 9 partially overlaps the text, and fig.10 is not marked .
Response 6:
thank you very much, we corrected them.
Reviewer 2 Report (New Reviewer)
Although the concept of study (non-surgical treatment of palatal fistula) is very interested. however, the were a several methodological mistakes. Some of them but not limited to
1. Introduction: introduction should be limited based on scope of paper, most of text introduction were not relevant. Authors should be more specific to palatal fistula directly. Authors did not present strong convincing palatal fistula .
2. I think this conservative non-invasive method could be succeed only for those fistulae with small size. So conclusion and results of this study was generalized. This is not true.
3. Methodology: there was no a real methodology (incomplete methodology), no clear inclusion criteria , no sample size calculation, no statistics etc
4. Authors did not follow up the patients radiographically and how authors assessing the outcomes of interest
5. There were a lot of typing and grammatical errors
6. Study did not consider size of fistula, in which non-surgical treatment could be recommended
7. Results presented as a case report.
8. Thus, based on the above comments, I suggest for authors to rewrite this manuscript as case reports with literature review.
Author Response
Response to Reviewer 2 Comments
- Introduction: introduction should be limited based on scope of paper, most of text introduction were not relevant. Authors should be more specific to palatal fistula directly. Authors did not present strong convincing palatal fistula
Response 1:
thank you very much, we corrected them.
- I think this conservative non-invasive method could be succeed only for those fistulae with small size. So conclusion and results of this study was generalized. This is not true.
Response 2:
Thanks a lot for the answer. it is necessary to increase the sample number in order to effectively define the dimensions as a statistically significant variable. We've edited the discussion and the results to match your observation.
- Methodology: there was no a real methodology (incomplete methodology), no clear inclusion criteria , no sample size calculation, no statistics etc
Response 3: we have better defined the inclusion criteria. Unfortunately, it is not possible to define a statistical calculation that would allow us to understand which variable increases the probability of success or otherwise of this technique. our goal was to propose a clinical technique. The goal is to design a statistically more valid study based on the clinical data collected.
- Authors did not follow up the patients radiographically and how authors assessing the outcomes of interest
Response 4:
thanks for the observation but it is not possible to share all the CT scans of the patients. furthermore, the CT scan only shows whether or not basal bone is involved. we thought it was enough just to describe which parameter had been considered.
- There were a lot of typing and grammatical errors
Response 5:
thank you very much, we corrected them.
- Study did not consider size of fistula, in which non-surgical treatment could be recommended
Response 6:
thank you very much, we corrected them.
- Results presented as a case report. Response 7:
thanks for the remark. we have created a summary table of the measurements of the various fistulas - Thus, based on the above comments, I suggest for authors to rewrite this manuscript as case reports with literature review.
Reviewer 3 Report (New Reviewer)
Dear Authors,
The article: 'Our school non-surgical approach for palatal fistula closure: a pilot study' was to propose a non-surgical approach for closing palatal fistulas (NSPF).
English language and style are fine.
Punctuation mistakes should be corrected.
Figure 1 should be bigger.
Sentences do not begin with numbers. This is an invalid notation eg. line 119 (47 patients were rehabilitated in our Head and Neck integrated care Department, in the complex operating unit of implantology and dental prosthetics, Department of Oral Sciences and Maxillo facial Surgery at Policlinico Umberto I, Sapienza, University of Rome (Rome, Italy).)
Also correct in line 157
Introduction is crealy written
Materials and methods:
Add information about the REC in this dection.
Highlight a section in materials and methods criteria for inclusion and exclusion from the study.
Correct the captions for figures and tables in accordance with the MDPI guidelines.
Prepare tables according to MDPI guidelines.
Add a table with abbreviations.
References should be prepared in accordance with the MDPI guidelines
To sum up, article should be reconsider after major revision.
Author Response
Response to Reviewer 3 Comments
Figure 1 should be bigger.
Response 1:
thank you very much, we corrected them.
Sentences do not begin with numbers. This is an invalid notation eg. line 119 (47 patients were rehabilitated in our Head and Neck integrated care Department, in the complex operating unit of implantology and dental prosthetics, Department of Oral Sciences and Maxillo facial Surgery at Policlinico Umberto I, Sapienza, University of Rome (Rome, Italy).
Also correct in line 157
Materials and methods:
Add information about the REC in this dection.
Highlight a section in materials and methods criteria for inclusion and exclusion from the study.
Correct the captions for figures and tables in accordance with the MDPI guidelines.
Response 2:
thank you very much, we corrected them.
Prepare tables according to MDPI guidelines.
Add a table with abbreviations.
References should be prepared in accordance with the MDPI guidelines
To sum up, article should be reconsider after major revision
Response 3:
thank you very much, we corrected them.
Reviewer 4 Report (New Reviewer)
The procedure described for closure of palatal fistulas is novel and the authors need to be complimented for the same. The manuscript requires lot of improvement to make it suitable for publication.
Title needs a change and may be changed to 'Evaluation of a novel technique for closure of small palatal fistula'.
Introduction should be brief and to the point and start with techniques of palatal fistula closure rather than classification of oral carcinoma. It should mention rationale for the proposed technique. It should finish with aim of the study.
Methodology needs lot of improvement. The authors should strictly describe the inclusion criteria. It is obvious from the figures of the representative cases shown the authors have selected narrow width fistulas. The technique may not be suitable for fistulas caused by partial or total maxillectomy defects. The authors should describe technique in detail and add a picture of the procedure. The methodology should be in past tense.
Result should clearly specify the success rate.
Conclusions should be drawn from the results of the study. Last line mentioning lasers should be deleted.
Self-citations in the reference should be deleted as these are not related to the study
Author Response
Response to Reviewer 4 Comments
Title needs a change and may be changed to 'Evaluation of a novel technique for closure of small palatal fistula'.
Response 1:
Thanks a lot for the answer. it is necessary to increase the sample number in order to effectively define the dimensions as a statistically significant variable. We've edited the discussion and the results to match your observation.
Introduction should be brief and to the point and start with techniques of palatal fistula closure rather than classification of oral carcinoma. It should mention rationale for the proposed technique. It should finish with aim of the study.
Response 2:
thank you very much, we corrected them.
Methodology needs lot of improvement. The authors should strictly describe the inclusion criteria. It is obvious from the figures of the representative cases shown the authors have selected narrow width fistulas. The technique may not be suitable for fistulas caused by partial or total maxillectomy defects. The authors should describe technique in detail and add a picture of the procedure. The methodology should be in past tense.
Response 3:
thank you very much, we corrected them.
Result should clearly specify the success rate.
Response 4:
thank you very much, we corrected them.
Conclusions should be drawn from the results of the study. Last line mentioning lasers should be deleted.
Response 5:
thank you very much, we corrected them.
Round 2
Reviewer 1 Report (New Reviewer)
Thank you for the revision of your paper. Unfortunately there are still many errors in the English language:
- one fistula two fistulae -derived from the Latin language (not fistulas)
- line 33 fistulae are continuous solutions (??)
- line 170 crucial instead of cruscial
- line 173 minimal instead of non surgical
-line 200 heel seems unusual - could probably be replaced by section ( to be answered by a native English speaking doctor)
-line 287 constituted around the fistula (see line 200)
- line 359 ff. treated instead of trated
- line 378 do you really mean a barrier and not a support ?
fig. 6, 9, 10 have no legend
Author Response
thank you so much for your kind review. We have made the changes also requested by other reviewers.Reviewer 2 Report (New Reviewer)
Unfortunately, authors did not address all my comments appropriately
Author Response
thank you so much for your kind review. We've edited the introduction by deleting unnecessary parts, making it shorter and more direct. we have underlined the importance of the size of the fistulas and summarized everything in a table. we have corrected the grammar errors. we have integrated the exclusion and inclusion criteria. We have added pictures of the probe used. To improve the communication of the technique we will add 2-3 images of the technique.Reviewer 3 Report (New Reviewer)
Punctuation mistakes should be corrected.
Sentences do not begin with numbers. This is an invalid notation eg. line 140 (47 patients were rehabilitated in our Head and Neck integrated care Department, in the complex operating unit of implantology and dental prosthetics, Department of Oral Sciences and Maxillo facial Surgery at Policlinico Umberto I, Sapienza, University of Rome (Rome, Italy).)
Also correct in line 217
Introduction is crealy written
Materials and methods:
Add information about the REC in this dection.
Highlight a section in materials and methods criteria for inclusion and exclusion from the study.
Correct the captions for figures and tables in accordance with the MDPI guidelines.
Prepare tables according to MDPI guidelines.
Add a table with abbreviations.
References should be prepared in accordance with the MDPI guidelines
To sum up, article should be reconsider after major revision.
Author Response
thank you so much for your kind review. We changed the text, the layout of the tables and images and added further changes proposed by other reviewers.Round 3
Reviewer 2 Report (New Reviewer)
Unfortunately, authors did not address my comments 2, 3, 4 and 8 appropriately
Reviewer 3 Report (New Reviewer)
Punctuation mistakes should be corrected.
References should be prepared in accordance with the MDPI guidelines
Artice can be accepted after corrections.
This manuscript is a resubmission of an earlier submission. The following is a list of the peer review reports and author responses from that submission.
Round 1
Reviewer 1 Report
Dear authors!
The outlined topic has the big practical significance especially for patients since such a treatment approach directly affects their quality of life.
Unfortunately, in the article evidence-based methods are not sufficiently and/or contradictory described. This is negatively affecting the scientific value of the material.
The main comments on the article can be found in the attached file.

Author Response
Good evening, thank you for the useful notes provided in order to improve the article. We have tried to modify following your advice. We await your kind reply
Reviewer 2 Report
Dear authors,
Thanks for presenting the minimally invasive technique to provide treatment for palatal fistulae. I have several question and suggestion:
1. Please clarify whether this is a case series, or a prospective clinical trial? If this is a clinical trial, then an IRB number should be provided.
2. Besides the classification/subgrouping, the author should also put the following information clearly into tables such as: the initial size of fistula, presence/abscess of maxillary bone involvement, the percentage of patients achieving full healing, mean+/- SD of residual defect size
3. Language needs to be uniformed. Palatal fistulae appeared as different names throughout the manuscript.
Author Response

(The authors gave the same response as above.)

Round 2
Reviewer 1 Report
Dear authors!
The quality of the second edition of the submitted manuscript has significantly improved. However, a number of questions and comments remain. Therefore, unfortunately, I cannot recommend this article for publication.
The comments on the article can be found in the attached file.
